



# On crustal composition of the Sardinia-Corsica continental block inferred from receiver functions

Fabio Cammarano[1], Henrique Berger Roisenberg[1], Alessio Conclave[1], Islam Fadel[2], and Mark van der Meijde[2]

[1]Department of Sciences, Roma Tre University, Largo san Leonardo Murialdo 1, 00146, Italy
[2]University of Twente, Enschede, The Netherlands

**Correspondence:** Fabio Cammarano (fabio.cammarano@uniroma3.it)

**Abstract.** Subduction-related geodynamic processes significantly influence plate tectonics and Earth's evolution, yet their impact on the continental crust remains poorly understood. We investigated the Sardinia-Corsica continental block, situated in the Mediterranean Sea, which has experienced intense subduction-driven geodynamic events. By analyzing P-wave receiver functions from our LiSard seismic network and publicly available stations, we aimed to understand crustal structure and

composition. We inferred the Moho depth and examined the P-wave to S-wave velocity ratio ($V_P/V_S$). We interpret our findings considering petrological data, heat flux measurements, and other geophysical information. We found that the Variscan granitoid batholith has the greatest Moho depths in both Sardinia and Corsica. $V_P/V_S$ ratios (ranging from 1.65 to 1.70) are consistent with average crustal values of $SiO_2$ between 65% and 70%. However, in central Corsica, two stations have exceptionally high $V_P/V_S$ values (>1.80), suggesting the possible presence of serpentinite throughout the crust. In Alpine Corsica, a station

exhibited similar high $V_P/V_S$ values but a shallower Moho depth of 21 km. The western part of Sardinia, where Cenozoic volcanism occurred, also showed a shallower Moho depth (20-25 km) and high $V_P/V_S$ values. The highest $V_P/V_S$ value (1.91) is recorded in an area where surface-wave dispersion curves from ambient noise identified the lowest average S-wave velocity and where the highest heat flux has been reported, indicating elevated crustal temperatures and possible presence of melt within the crust. Overall, our results indicate that the recent geodynamic processes have left the granitoid batholith almost

intact, with minimal alteration to its composition.

## 1 Introduction

The Sardinia-Corsica continental block serves as a natural laboratory for understanding the shaping of the continental crust by geodynamic processes occurring at plate boundaries. The primary constituent of the Sardinia-Corsica 'micro-continent' is a granitoid batholith, that formed as a consequence of the Variscan (or Hercynian) orogenesis. This batholith spans approximately

15.000 $km^2$, encompassing nearly the entire Corsica and a significant portion of East Sardinia (Fig. 1). There exists a petrological continuity between this batholith and those found in Provence (Elter et al., 2004). The northeastern region of Corsica, known as Alpine Corsica, represents a segment of the Alpine orogenic system. It comprises a series of tectonic units derived from both continental and oceanic sources, which have undergone varying degrees of metamorphism (Malavieille et al., 1998;



Vitale Brovarone et al., 2013). Abundant high-pressure lawsonite-bearing rocks are found in this region, including lawsonite-
bearing eclogite rocks that have been exhumed from a large depth of approximately 70 km, making them exceptionally rare on
a global scale (Vitale Brovarone et al., 2013).

Petrological and structural data indicates that the Hercynian basement of the Corsica island experienced deformation during
the dynamics of the Alpine orogeny. For instance, investigations utilizing low-temperature thermochronology and Raman
spectroscopy of carbonaceous materials have provided insights into the maximum temperatures attained by Corsican rocks
during the Alpine tectonic events (Rossetti et al., 2023). These findings suggest that the Alpine orogenic wedge and/or foreland
basins likely covered almost the entire Corsican island.

Approximately 35 million years ago until around 10 million years ago, the Apennine-Tyrrhenian slab underwent asymmetric
retreat, resulting in the opening of the Ligurian-Provencal back-arc basin (e.g., Faccenna et al., 2014). During this period,
the Sardinia-Corsica block separated from mainland Europe and underwent rotation. Within this period, there was abundant
volcanism related to subduction occurring in the western region of Sardinia. Additionally, there were occurrences of less
abundant and geochemically distinct volcanism in more recent times (see Lustrino et al., 2013, for a thorough review of
Cenozoic volcanism in Sardinia). The rapid opening of the Tyrrhenian Sea, probably initiated around 5 million years ago, can
be attributed to the rapid retreat of the oceanic-lithospheric portion of the slab, likely caused by the completion of a slab tearing
(Faccenna et al., 2014, and references therein). As a consequence of this process, the Sardinia-Corsica block remained behind
as the Tyrrhenian Sea expanded.

Significant evidence of the relatively recent and intense geodynamic evolution can be observed in both the Cenozoic and
more recent volcanic deposits, as well as in the pronounced topography that characterizes both Corsica and Sardinia. Fur-
thermore, the heat flow in these islands is relatively high compared to other regions. A study by Della Vedova et al. (1995)
indicates that Corsica has an average heat flow of around 75 $mW/m^2$, which is higher than the typical values observed in
other parts of the Hercynian basement. For instance, in Provence, heat flow values of approximately 60 $mW/m^2$ are observed.
In Sardinia, the heat flow values associated with the Variscan batholith are generally less pronounced and similar to those
observed in Provence. However, in the western portion of Sardinia, notably in the Campidano graben, even higher heat flow
values exceeding 100 $mW/m^2$ have been recorded (Della Vedova et al., 1995). Additionally, both islands exhibit abundant
thermal springs. Some of the hottest springs in Sardinia are located along the margin of the granitoid Variscan batholith and
are likely linked to the lateral flow of water through the radiogenic-rich granite (Cuccuru et al., 2015).

Deep seismic refraction profiles have revealed a relatively uniform Moho depth in both Corsica and Sardinia, with slightly
larger values in the central regions of the islands. The maximum crustal thickness recorded is approximately 33 km in Corsica
and 34 km in Sardinia (Egger et al., 1988). However, as we move towards the coastal areas, the Moho depth shows a decreasing
trend, with values ranging between 30 and 23 km. The average P-wave velocities ($V_P$) in these areas are around 6.4 $km/s$ and
6.3 $km/s$, respectively. These findings are supported by the gravimetric Bouguer anomaly map, as reported by Carrozzo et al.
(1997), which is consistent with the observed crustal thicknesses.

A receiver function analysis has been conducted, for the first time, by Megna and Morelli (1994), utilizing 20 teleseismic
events recorded at the VSL station in South Sardinia. The results of Megna and Morelli were subsequently reviewed and





confirmed by van der Meijde et al. (2003). These investigations estimated a Moho depth of 29 ± 1 km at the VSL station in
South Sardinia. This station is also included in the present study.

In 2015, Afilhado et al. conducted an investigation of the Sardinia margin, encompassing both the sea and mainland, using
wide-angle and reflection seismic data. They determined a crustal thickness of 27 km on land, and identified two distinct layers,
measuring 12 km and 15 km in thickness, with corresponding P-wave velocities ($V_P$) ranging from 5.9-6.4 km/s and 6.5-6.7
km/s, respectively. At the Moho discontinuity, a jump to velocities of 7.9-8.0 km/s was inferred.

In June 2016, the LiSard (Lithosphere of Sardinia) project was initiated with the goal of studying the crustal and lithospheric
mantle structure of Sardinia. The project was motivated by the limited seismic coverage in Sardinia, as the region is relatively
seismically inactive, resulting in a lack of robust constraints on the deep lithospheric structure. Additionally, the strategic
location of having an array in the center of the Mediterranean facilitated large-scale tomographic studies (Magrini et al.,
2022; Agius et al., 2022). The LiSard project consists of a network of 10 seismic broadband stations uniformly distributed
throughout the territory. The data acquisition, spanning over two years, has generated a significant amount of data, which has
only been partially analyzed thus far. For example, the analysis of ambient noise cross-correlation between pairs of stations has
allowed for the measurement of dispersion curves of surface waves (Rayleigh and Love waves), enabling the development of
a tomography model of shear waves ($V_S$) (Magrini et al., 2019). The $V_S$ (shear wave velocity) model derived from the LiSard
project reveals a distinct division in the upper crust, with lower $V_S$ values observed in the western portion and higher $V_S$ values
in the eastern portion. Although the study only partially covers Corsica, the central region of Corsica also exhibits elevated
$V_S$ values, similar to those found in the eastern part of Sardinia. This suggests a pattern of higher shear wave velocities in the
central and eastern parts of the study area, while the western regions display comparatively lower Vs values.

In this study, we conducted an analysis of P-receiver functions using data from 11 locations of the LiSard project and 10 pub-
lic stations from the INGV, RESIF, and MEDNET networks. Our main objective was to obtain a more accurate determination
of the Moho depth and gain insights into the crustal composition by examining the retrieved $V_P/V_S$ ratios.

To interpret our results, we integrated them with previous seismic investigations and considered aspects such as crustal
composition and temperature. In order to achieve this, we performed thermodynamic modeling and incorporated constraints
from petrology and heat-flow studies. By combining these different lines of evidence, we aimed to improve our understanding
of the crustal structure, its composition, and the thermal conditions in the study area.

**2    Data and Methods**

**2.1    The P-wave receiver function**

The P-wave receiver function (RF) analysis is a widely-used technique employed in the investigation of the subsurface structure
of the Earth, with a specific focus on discerning boundaries between different geological layers (e.g., Langston, 1977; Vinnik,
1977; Ammon, 1991; Zandt and Ammon, 1995; Piana Agostinetti and Amato, 2009). This method involves the examination
of the seismic response to teleseismic P-waves, which undergo conversions to S-waves at those boundaries, thereby yielding
crucial information about the properties of seismic discontinuities beneath the receiver. For instance, when a teleseismic P-





wave encounters the Moho, the interface between the Earth's crust and mantle, it generates a converted shear-vertical (SV) wave known as the Ps phase. By accurately measuring the time delay between the P-wave and the generated S-wave, it becomes possible to estimate the depth to the boundary and deduce relevant details concerning the velocity ratio between P-wave and S-wave velocity ($V_P/V_S$) within the crust. In addition to the Ps phase, other seismic phases, including PpPs, PsPs, and PpSs, arise due to multiple reflections. The incorporation of these phases in the receiver function analysis contributes to constraining the determination of velocities, in particularly the $V_P$ to $V_S$ ratio, and the accurate assessment of depths pertaining to subsurface discontinuities.

The P-wave receiver function is obtained by assuming plane-wave approximation and through a deconvolution process applied to the vertical and radial components of the seismogram. This deconvolution effectively removes the common elements present in the waveforms, which include information about the seismic source and the travel path. Several deconvolution techniques, either in time or frequency domain, have been proposed. In this study, we employ the classical time-domain iterative deconvolution. This method was originally introduced by Kikuchi and Kanamori (1982) for estimating the source time function of large earthquakes. Subsequently, Ligorria and Ammon (1999) applied this method to determine the receiver function for teleseismic earthquakes.

The iterative deconvolution approach consists in finding the best fit between the observed horizontal seismogram and a predicted signal. This is done by convolving an iteratively updated spike train with the vertical-component seismogram. The process starts by cross-correlating the vertical and radial components to determine the lag of the first and largest spike in the receiver function. The optimal time corresponds to the peak with the highest amplitude in the cross-correlation signal. Next, the current estimate of the receiver function is convolved with the vertical-component seismogram and subtracted from the radial-component seismogram. This step is repeated to estimate the lags and amplitudes of additional spikes in the receiver function. With each iteration, the misfit between the convolved vertical seismogram and the radial seismogram is reduced as more spikes are incorporated into the receiver function. The iteration process continues until the reduction in misfit with each additional spike becomes insignificant, indicating that further refinement is no longer significant. The approach can be also applied to transverse motion.

In the case of high-quality data, the selection of alternative deconvolution methods has minimal impact on the resulting receiver function. We conducted test for a limited set of data and specifically applied the water-level deconvolution method (Clayton and Wiggins, 1976), which is another commonly used technique for extracting receiver functions. We observe only minor deviations in the obtained receiver functions.

Like any inverse process, each deconvolution method requires subjective decisions regarding regularization terms. In practice, a balance must be struck between the complexity of the model and the reduction of data variance. In the case of the time-domain iterative deconvolution, the key parameter is the Gaussian width factor. This factor controls the bandwidth of the signal, with larger values corresponding to wider bandwidths (Ligorria and Ammon, 1999). In general, a wider pulse provides lower time resolution but may be more stable in the presence of noise. On the other hand, a narrower pulse provides higher time resolution but may be more sensitive to noise. The choice of the width of the Gaussian pulse depends on the specific





characteristics of the seismic data, including the signal-to-noise ratio and the desired level of detail in the resulting receiver function.

### 2.1.1 Thermodynamical modeling and reference lithologies

Physical properties, such as density and seismic velocities ($V_P$ and $V_S$) are governed by temperature and composition. Thermodynamical modeling is a valuable tool for studying these properties, and accounts for mineralogical phase transitions. While metastable rocks may exist in the Earth's crust, previous research (Guerri et al., 2015; Diaferia and Cammarano, 2017) demonstrated the applicability of this modeling approach for crustal properties. In this study, we follow the same procedure as Diaferia and Cammarano (2017), using the $Perple\_X$ software (Connolly, 2005) with a specialized database specifically designed for computing seismic velocities within the crust (for detailed information, refer to Diaferia and Cammarano, 2017). The primary advantage of using a single thermodynamic database of mineral species is the ability to compute consistent bulk-rock properties (seismic velocities and density) for various chemical compositions. However, it is important to note that this method does not account for metastability and has certain limitations, such as the exclusion of porosity, which can be significant in the uppermost portion of the upper crust. Additionally, the method only considers isotropic properties.

The thermodynamical modeling has been applied to model the properties of several compositions, which we deemed to be representative of the Sardinian-Corsica outcropping rocks (Table 1). We divided the block into three major provinces. The first province is characterized by the presence of a Variscan (300 million years old) granitoid batholith in the northeastern part of Sardinia and Corsica, as well as Variscan metamorphic rocks in the southeastern part of Sardinia (Figure 1). The most abundant rocks in this province are granodiorites and monzogranites (Poli et al., 1989), which are rich in quartz and therefore have high $SiO_2$ content in their chemical compositions. We also model a global granite composition (Blatt and Tracy, 1996). In the western part of Sardinia, on the other hand, there are volcanic rocks that are related to the subduction of the African plate and were emplaced between 30 million years ago and 10 million years ago (Lustrino et al., 2013). These rocks have a more mafic composition. Additionally, there is more recent magmatism characterized by compositional variations, which Lustrino et al. (2013) divided into two groups of volcanism: RPV (Radiogenic Pb Volcanism) and UPV (Unradiogenic Pb Volcanism) (only RPV is shown in the Table 1 as UPV is less abundant). Finally, the northeastern part of Corsica, known as Corsica Alpina, exhibits, as mentioned earlier, metamorphic rocks at various grades (blue and green schists), and basaltic and metabasaltic compositions with the presence of ophiolitic sequences and mantle serpentinites. A globally rare rock, but abundant in Corsica Alpina, is an eclogite containing Lawsonite (Vitale Brovarone et al., 2011), shown in Table 1 and included in our tests.

Furthermore, we modeled the globally averaged chemical compositions of the upper, middle, and lower crust, as proposed by Rudnick and Gao (2014). This was done to illustrate the implications of transitioning from a silica-rich (sialic) upper crust to a more magnesium- and iron-rich (mafic) composition, and to have a reference with respect to the Sardinia-Corsica compositions.

The thermodynamical modeling has been performed employing a seven-oxides system: $SiO_2 - Al_2O_3 - FeO - MgO - CaO - Na_2O - K_2O$. Additionally, a small amount of $H_2O$ (0.25 wt%) was incorporated, which indirectly affects the $V_P/V_S$ ratio by reducing the solidus temperature and aiding in the formation of a more realistic mineral assemblage (Guerri et al.,



2015). It is important to emphasize that our main objective was not to precisely replicate the exact mineral compositions found in nature. Rather, we aimed to estimate seismic velocities and compare relative variations among different compositions. In relation to this, we also investigated the influence of silica content in the granitoid batholith on the $V_P/V_S$ ratio. To accomplish this, we employed the global reference upper-crust composition proposed by Rudnick and Gao (2014) and adjusted the $SiO_2$ content to various levels (as presented in Table 2).

**2.2 Datasets and processing**

In our study, we utilized both the available broadband public seismic stations and our in-house LiSard seismic network to compute P-wave receiver functions. The locations of the seismic stations analyzed in this study are depicted in Figure 1. The LiSard network, operational from June 30th, 2016, to September 15th, 2018, consisted of 10 broadband seismic stations equipped with Nanometrics Trillium posthole sensors.

One of the LiSard stations, UT.001, was initially located in La Maddalena, North Sardinia for a period of 14 months before being relocated to the Campidano graben as station UT.011. The remaining stations operated continuously for over two years, recording more than 200 earthquakes with magnitudes between 5.5 and 8.0 and epicentral distances ranging from 25 to 95 degrees. The earthquake catalog used for our analysis is sourced from IRIS (Incorporated Research Institutions for Seismology).

For the 11 public stations examined in this study, we considered an earthquake catalog spanning from June 30th, 2016, to 175 January 1st, 2023. Some stations, such as FR.MORSI, were active throughout the entire time interval, enabling the recording of a significant number of teleseismic events.

We use python open-source libraries for the analysis of the waveforms. We extensively used obspy routines (Beyreuther et al. 2010; https://docs.obspy.org/index.html) and we use seispy (Xu and He, 2022; https://seispy.xumijian.me/index.html) for the processing and post-processing of the receiver functions.

In our data processing workflow, we extracted P-waveforms from the three-component seismograms, spanning from 10 seconds before to 120 seconds after the theoretical P arrival time, which has been inferred using $TauP$ (Crotwell et al., 1999) and the $AK135$ model (Kennett et al., 1995). Prior to analysis, we applied preprocessing steps, including the removal of the mean and trend from the waveforms and the application of a bandpass filter between 0.05 and 2 Hz.

We rotated the North (N) and East (E) components of the seismic waveforms to obtain the radial (R) and transverse (T) 185 components, and we thus computed the P-wave RF. As mentioned earlier, we utilized the iterative deconvolution method. Following extensive testing, we decided to employ a Gaussian width factor of 1.5. This choice is conservative and aligns with our focus on determining the depth of the Moho, which is the primary seismic discontinuity typically imaged in the P-wave coda across most of our study locations.

The receiver functions from different teleseismic events at one single station can be, sometimes, different. These differences 190 can be attributed to variations in the source characteristics, propagation paths, and station-specific effects that are not fully removed by the deconvolution process. Therefore, it is important to select the best quality data. This is done by an automatic quality control, that is, therefore, an important part of the process. There are two criterion to remove bad-quality receiver functions. The first one is based on the signal-to-noise ratio of the seismic waveform. We employ a value of 1 dB. The second





one is based on a root-mean-square threshold between the original and recovered receiver function, which we choose to be 0.25.
Both parameters were selected based on tests with various stations. In our opinion, these values are able to retain a statistically
significant amount of data with relatively good quality. In addition, however, a manual inspection of the RFs that passed the
selection criteria has been also carried out. In particular, we remove few waveforms that do not align with others on the first
arrival. These steps help to ensure a consistent and reliable dataset for subsequent analysis.

In Table 3 we report a list of the teleseismic P-wave RFs analyzed at each station and the number remaining after quality
control procedures. It is noteworthy that the number of discarded RFs varies greatly among different locations. We have a num-
ber of events ranging from 200 to 230 for the nine LiSard locations which have been active for two years. With the exception
of UT.002 and UT.005, all other seven stations have more than 90 RFs that meet the quality criteria. UT.011, which has been
recording for approximately nine months, captured 87 events, but only 14 of them were deemed suitable for further processing.
This station is located in the Campidano Graben, an area characterized by a thick and structurally complex sedimentary cover
that can introduce reverberations and complicate the waveforms. UT.001, situated in North Sardinia, exhibited significant noise,
resulting in the selection of only a small fraction of the analyzed RFs after quality control. In addition, the few remaining RFs
do not present a consistent pattern. As a result, it was decided to exclude this station from further processing. Comparatively,
some public stations have been operational for a longer duration, resulting in a higher number of recorded teleseismic events,
as indicated in Table 1. The number of the eventually selected RFs is variable also in this case. Notably, IV.AGLI station has
only 32 selected RFs available for further analysis.

## 2.3 H-k stacking analysis

The H-k stacking method, introduced by Zhu and H. (2000) is a simple, but effective method to determine the crustal thickness
and the $V_P/V_S$ ratio beneath each station. The general idea is to exploit not only the main Ps for the stacking but also the major
multiples, i.e. the PpPs and the PpSs+PsP. The time delay of the solely Ps phase from the P arrival can be used to estimate the
thickness of the crust, but trades off strongly with the $V_P/V_S$ ratio: a variation of 0.1 in the latter causes about 4km change in
the crustal thickness. A slightly sensitivity on $V_P$ is also occurring: i.e. a 0.1 km/s variation gives a 0.5 km change in the crustal
thickness. To mitigate this ambiguity, it is possible to consider the PpPs and the PpSs+PsP phases. According to the different
move out of the phases, the three phases will coherently stack only when the correct crustal thickness (named H) and $V_P/V_S$
(named k) ratios are used.

Using several RFs for the same H-k stack increases the signal-to-noise ratio. However, it is noteworthy to remember that,
owing to the presence of 3-D structural complexity within the crust and/or at the Moho depth, the multiples can be partly
masked. The weighting scheme of each of the three seismic phases therefore, has to be carefully checked for each station. For
the sake of simplicity, we adopt, in this stage, the same weighting scheme for each station, that is 0.6 for the Ps phase and
0.3 and 0.1 for the PpPs and PsPs+PpSs, respectively. Later, we will discuss specific cases for which we had to modify the
weightings.

The search has been carried out by using values of Moho depth between 20 and 50 km and of $V_P/V_S$ between 1.55 and
1.90. We assume a value of $V_P$ equal to 6.4 km/s. Note that the error bars resulting from the H-k stacking are based on the 95%





confidence interval (2 $\sigma$). These are only strictly valid if the simplifying assumptions of homogenous, isotropic and horizontal layers are valid. Therefore, they represent an optimistic value.

## 3 Results


In order to ensure high-quality results, the stacked P-wave receiver function requires a sufficient number of reliable receiver functions with a good azimuthal coverage. In an ideal scenario, where the subsurface consists of a simple layered structure without dipping interfaces, anisotropy, or other three-dimensional structural heterogeneity, all the energy related to the converted P-to-S waves would be observed solely on the radial component, specifically would be only made SV (S-wave in the

vertical plane) energy. However, real-world situations are more complex and deviate from this ideal case. Hence, it is crucial to compute the transverse receiver functions alongside the radial ones. The transverse receiver functions provide additional information that aids in the interpretation of observations.

Figure 2, top panel, illustrates the radial and transverse receiver functions obtained from UT.003, one of the nine LiSard stations that recorded seismic events over a two-year period. The events are arranged based on their back-azimuth. Similarly

to the other LiSard stations, UT.003 exhibits a consistent pattern among the various events. The only exception being UT.011 (Figure 2, bottom panel), which has a limited number of records satisfying the quality criteria adopted, and even those show a noisy pattern.

For the public stations, we present the receiver functions from the extensively studied MN.VSL station in Figure 3, top panel. The receiver functions from this station demonstrate a good azimuthal coverage, indicating a comprehensive dataset for

analysis. Additionally, we include the IV.AGLI station (Figure 3, bottom panel), for which only a limited number of the total receiver functions meets the quality criteria (see Table 3). Also in this case, the selected RFs exhibit significant noise in the waveforms, compromising their reliability.

The computed radial and transverse receiver functions for all the other analyzed stations are shown in the supporting material (Figure S1 to S17). In addition, readers can find in supporting material the links to the original three-component waveform data

selected for each event and the full procedure for reproducing all our results. This allows readers to access and examine the complete set of receiver functions obtained from the study and ensures reproducibility of our results.

The obtained H-k estimate for all the LiSard stations, accompanied by the 95% confidence interval, are presented (Figure 4 and 5). The output of the Zhu and Kanamori's (2000) technique is displayed in the bottom right panels of each figure. The stack values for each individual phase are presented in the remaining panels. Maximum stack values are attained when all three

phases arrive coherently. Normalized values ranging from 0 to 1 are employed in each panel.

The arrival times are well-defined for the various expected phases, particularly for the Ps phase, in most of the LiSard stations, indicating that these receivers can be considered "high quality". The high quality of our data is reflected in coherent images of the individual seismic phases (Ps, PpPs and the PpSs+PsPs) and well-constrained Moho depths and $V_P/V_S$ ratios. Only few stations show some peculiar features that pose challenges for a clear interpretation.



We show in Figure 4 the H-k stack of 4 Lisard stations characterized by the most intricate features, i.e. UT.011, UT.009, UT.005 and UT.006.

For stations UT.001 and UT.011, a low number of receiver functions were obtained (11 and 14, respectively, as shown in Table 3). In the case of UT.001, this limited number of receiver functions did not allow for a reliable estimation of H-k parameters due to the short recording interval and incoherent waveform traces. However, for the UT.011 (Villafranca) station

(shown in top-left panel of Figure 4), a Moho depth of 30.8±2.13 km and a Vp/Vs ratio of 1.66±0.12 were estimated.

Station UT.009, situated at Asinara's island in northwestern Sardinia (as shown in Fig. 1) lacks a prominent Ps arrival (Figure 4, top-right panel). Upon examining the RFs, we observe a negative polarity signal in most of the RFs, that may be related with the presence of a low-velocity layer.

Similarly, stations UT.005 and UT.006 (bottom panels of Figure 4) are characterized by complexities that cannot be attributed

to a single discontinuity. The Ps is largely visible, but the multiple phases are partially masked by other signals. To account for these complexities, we apply a different weighting scheme (0.8, 0.1, 0.1 for Ps,PpPs and the PpSs+PsP, respectively ) for these two stations. This weighting scheme ensures a more accurate mean, albeit with higher uncertainties due to the increased trade-off between H and k. The same applies to UT.006, although the variation resulting from the new weighting scheme is negligible.

For the remaining six stations, shown in Figure 5, the high quality of our data is reflected in coherent images of the individual seismic phases (Ps, PpPs and the PpSs+PsPs) and well-constrained Moho depths and $V_P/V_S$ ratios.

Station UT.008 (bottom-left panel in Figure 5) exhibits the shallowest Moho depth and high Vp/Vs features, which align with the station's location predominantly sampling the oceanic environment. This region has been previously identified as having the lowest Vs within the crust, as indicated by a previous study by Magrini et al. (2019) based on surface waves from

ambient noise. It is worth noting that for station UT.008, we had to expand our H-k search to accurately compute the confidence interval, requiring a k value of 2.0. Conversely, for station UT.009, the search had to be extended to k = 1.5.

The shallow Moho depth at station UT.007 (at Carloforte) and a relatively higher $V_P/V_S$ ratio was also expected.

The receiver functions from both the French (FR) and Italian (IV) public networks exhibit excellent quality as well. The FR.AJAC (Figure 6, top-left) displays blurred multiples, which could be attributed to crustal heterogeneities. The estimated

crustal thickness (H) for this station is 25.4±1.10 km, with a corresponding $V_P/V_S$ ratio (k) of 1.82±0.05. The FR.MORSI station (Figure 6, top-right) exhibits a much simpler structure, characterized by a shallow crust with a thickness of 21.4±0.81 km and a higher $V_P/V_S$ ratio of 1.85±0.06. These results align with expectations, considering the station's location in the northernmost part of the Alpine Corsica (see also geological map in Figure 1). Noteworthy findings were obtained for the FR.CORF (Corte) and FR.SMPL (Barrage de Sampolo) stations (bottom panels of Figure 6), revealing a relatively deeper

Moho depth. The estimated crustal thickness is 32.5±1.69 km for CORF and 33.9±0.07 km for SMPL, indicating a substantial Moho depth for both locations. These findings are consistent with previous studies employing alternative methodologies, lending support to the robustness of the results. Remarkably, an unexpectedly high $V_P/V_S$ ratio was observed in these two stations. Specifically, the $V_P/V_S$ ratio was determined to be 1.81±0.076 for CORF and 1.88±0.001 for SMPL (Figures 6,





bottom panels). Additionally, it is worth noting that the CORF station exhibits an indistinct Ps phase, implying the presence of

intricate wave propagation characteristics specific to the Central Corsica region.

The CELB station, part of the INGV network and situated on the Island of Elba, exhibits an interesting characteristic in terms of the $V_P/V_S$ ratio (Figure 7, top-left panel). The ratio, determined as k=1.70±0.06, indicates a relatively low value compared to what expected for oceanic environment. This observation highlights the significant influence of the region's outcropping granodiorite rocks on the seismic properties. Additionally, the estimated Moho depth for this station is H=22.7±1.1 km.

Moving further south, the Aglientu station (IV.AGLI) encountered challenges during the processing of receiver functions due to the presence of considerable noise in the data. Consequently, a significant number of tracks were discarded based on the adopted quality criteria. The remaining receiver functions exhibited a heterogeneous trend as well. In the subsequent H-k analysis (Figure 7, top-right panel), a different weighting scheme was applied for this station, namely 0.7 for Ps, and 0.2, 0.1 for the PpPs and the PpSs+PsP, respectively. Despite these adjustments, the trade-off between crustal thickness and $V_P/V_S$

ratio persisted, resulting in larger uncertainties in the estimates. We obtain a Moho depth of 23.4±7.51 km and a $V_P/V_S$ ratio of 1.84±0.22. The incorporation of multiples did not effectively reduce the trade-off between these two parameters.

The MN.SENA station, situated at a depth of 111m to enable precise monitoring of regional seismic noise (Di Giovanni et al., 2020), has been operational since 2019. The recorded waveforms at this station are characterized by notable complexity. Figure 7, mid-left panel, reveals a blurred Ps phase, accompanied by subsequent multiple arrivals. The intricate waveform

characteristics observed during the analysis of receiver functions likely arise from pronounced three-dimensional structural heterogeneity or an unclear transitional zone.

The IV.DGI (Dorgali) station shares similar characteristics with its neighboring SENA station. In the case of DGI, a substantial number of receiver functions had to be excluded based on the selection criteria applied during the receiver function analysis. After conducting thorough examinations, it was determined that utilizing only the Ps phase would be the most appro-

priate course of action, despite the presence of ambiguous multiples. Consequently, the advantage offered by the H-k method, which considers multiple phases jointly, was forfeited. Nevertheless, coherent values were obtained that effectively represent the constraints imposed by this specific station (Figure 7, mid-right panel).

The MN.VSL station in Villasalto (Figure 7, bottom-left panel), which has been extensively studied over the years, yielded excellent data through the initial processing phase. The coherence observed between the Ps phase and multiples enabled the

successful application of the H-k technique. The resulting values of H=33.0±1.3 and k=1.67±0.04 were found to be consistent with previous studies conducted on this particular station.

Finally, the nearby IV.CGL has a limited number of receiver functions that have passed the quality check, and shows scattered waveforms, such as to obtain two localized minima with H-k stacking, one centered at H=27.4 km and k= 1.85 and the second at H=35.9±1.64 km and k=1.63±0.07, respectively. Furthermore, the Ps phase, as for VSL, is relatively weak in amplitude,

consistent with previous results (Pondrelli et al., 2019).

The results obtained for each station are reported in Table 4. Together with the estimated Moho depths and $V_P/V_S$ ratio and respective confidence intervals, we also report the average $V_S$ for the crustal depth inferred from the Magrini et al. (2019) model and its uncertainty in percentage. Finally, last column is the Moho depth estimated with a fixed $V_P/V_S = 1.7$.





## 4 Discussion and interpretation

Receiver functions, unlike previous deep refraction profiles, are not very sensitive to lateral velocity variations. A nominal horizontal resolution for the crust-mantle discontinuity is on the order of 20-30 km. Dipping interfaces, complex structural heterogeneity and anisotropy introduce complexities to the obtained RFs and complicate interpretation. Modeling and interpretation of anisotropic effects in the RFs will be topic for future work. We note here that the azimuthal coverage of our data was appropriate for conducting a reliable isotropic analysis. The results obtained from our receiver-functions analysis provide,

therefore, unprecedented resolution on the depth of the Moho and offer an average value of the $V_P/V_S$ ratio (Figure 8), which is a crucial parameter for determining the average composition of the crust (Christensen and Mooney, 1995). These findings can be interpreted in terms of petrological, thermal, and geophysical constraints.

As previously mentioned, we considered three major petrological provinces, and we have conducted thermodynamical modeling to determine the physical properties of representative lithologies in a state of equilibrium as a function of pressure (depth)

and temperature. The chemical compositions of these lithologies can be found in Table 1. Additionally, we have computed, the P-T properties of the upper-, middle- and lower-crust from the global compilation of Rudnick and Gao (Table 2).

In Figure 9, we present the results of our thermodynamical modeling in terms of $V_P/V_S$ ratio as a function of depth (we convert pressure to depth using the PREM pressure profile, Dziewonski and Anderson, 1981). We also include reference thermal profiles based on steady-state calculations of continental geotherms. Specifically, we show a geotherm that has been

computed with a surface heat-flow of 60 $mW/m^2$, along with two additional geotherms corresponding to heat flows of 40 and 80 $mW/m^2$.

Once the surface heat flow is known, it is possible to estimate temperatures at greater depths by considering a radiogenic heat distribution in the shallow crust and taking into account the thermal conductivity of rocks (e.g., Chapman, 1986). For the geotherms displayed in Figure 9, we assume a thermal conductivity value of 2.5 $Wm^{-1}K^{-1}$ and a fixed radiogenic heat

production ($A_C$= 1.0 $\mu Wm^{-3}$). These values are chosen arbitrarily, since the scope of plotting the steady-state continental geotherms is only to provide a reference range of possible temperatures within the crust. Indeed, it is important to note that due to the recent geodynamic processes that Sardinia and Corsica have undergone (as described in the introduction), the assumption of steady-state conditions may not be applicable to our study region. In any case, we point out that the radiogenic heat value of Variscan batholith in Sardinia and Corsica has been estimated only slightly higher than the global average for the upper crust

(Verdoya et al., 1998; Puccini et al., 2014).

In general, the crystalline crust consists of a sialic upper-crust, an intermediate middle crust and a mafic lower crust. As shown in the panels representing the reference crustal composition (Figure 9, left panels), the values of the $V_P/V_S$ ratio increase, but slightly, with depth within the crust. The addition of $H_2O$, which is essential to obtain a more realistic mineralogy, has its main effect owing to the reduction of the solidus. Anelastic effects at solid state, here not considered, could be also

enhanced. The most evident variations at varying depth (pressure) and temperature are related to the break-down of plagiocase and to $\alpha$ to $\beta$ quartz transition (Diaferia and Cammarano, 2017).



By examining the right panels in Figure 9, it becomes apparent that the rocks representing the Variscan batholith in Sardinia and Corsica are characterized by the lowest values of the $V_P/V_S$ ratios. Along the reference geotherm at $60\ mW/m^2$, a typical monzogranite has a $V_P/V_S$ equal to $\sim$1.65. Slightly higher values, but not exceeding 1.7, are estimated for a granodiorite

and for a globally average granite composition. Conversely, the volcanic rocks outcropping in western Sardinia, as well as the metabasalts and metamorphic rocks of Alpine Corsica, are characterized by higher values at same conditions. Specifically, we have calculated $V_P/V_S$ values approximately of 1.75 for an average chemical composition of the Cenozoic, subduction-related volcanic rocks, and $\sim$1.8 for a Lawsonite-Eclogite (Figure 9).

The stations located on the Sardinian portion of the batholith exhibit relatively low values of the $V_P/V_S$ ratio, which is

consistent with a silica-rich composition of the crust (Figure 8).

As mentioned in the introduction, the study of the ambient noise using the LiSard network has permitted to extract and model dispersion curves of surface waves (Magrini et al., 2019). Although these waves do not provide precise information on sharp discontinuities, they offer a reliable constraint on the average shear-wave velocity of the crust. The average $V_S$ values for a fixed 30 km crust, as determined by Magrini et al. (2019), are presented in Figure 10. Additionally, Table 4 provides

the average $V_S$ values estimated for each station. The shear-wave model covers a large portion of the study area, and reveals an evident east-west trend in Sardinia, with relatively lower average $V_S$ in the eastern part. Corsica is only partially resolved. However, the central area shows comparable velocities to eastern Sardinia. It is important to note that the absolute velocities derived by surface-wave based models are partly influenced by regularization. On this regard, a $V_S$ scale with larger velocity variations could be also consistent with the surface-wave data.

To quantitatively estimate the average silica content of the Sardinian crust, we model how the physical properties change as a function of $SiO_2$ (Figure 11, see chemical compositions in Table 2). The variation in silica content has a relatively modest impact on the average shear-wave velocity ($<V_S>$). For instance, along a geotherm based on a surface heat flow of 60 $mW/m^2$, $<V_S>$ increases slightly from $\sim$3.75 to 3.80 as the $SiO_2$ content varies from 60 to 70 wt% (Figure 11). In contrast, $V_P$ decreases from $\sim$6.6 to 6.4 within the same range. The combined ratio $V_P/V_S$ shows the most significant variations,

ranging from 1.76 to 1.65.

Despite the inherent uncertainties in our estimates, as well as in other geophysical and thermal constraints, the thermodynamical modeling gives valuable insights for the potential range of average silica content in the Sardinian Variscan crust. In Figure 11, we identify a specific region that is consistent with both the thermal and geophysical constraints. This region corresponds to a $V_P/V_S$ ratio ranging from 1.65 to 1.70, consistently observed in eastern Sardinia (refer to Figure 8). The average

shear-wave velocity ($V_S$), which is relatively higher in eastern Sardinia (as shown in Figure 10), has values around 3.7 km/s. Finally, the $V_P$ is on the order of 6.4 km/s (Egger et al., 1988). Taking these different findings, we can infer that the silica content of the Sardinian Variscan batholith falls within the range of 65 to 70%.

It is interesting to note that the $V_P/V_S$ is less sensitive to temperature than absolute velocities ($V_S$ and $V_P$), as shown in Figure 11, except when temperature reaches levels high enough to approach the solidus. However, colder thermal structures

can be confidently excluded based on multiple factors. Firstly, the measured heat flow indicates that colder thermal structures



are unlikely. Additionally, colder structures would result in excessively high absolute velocities, providing further evidence against their presence in the region.

Our modelling is also able to establish a consistent relationship between seismic velocities and density as a function of composition and temperatures within the crust. For instance, assuming a 30 km thick crust composed of silica-variable compositions
ranging from 65 to 70%, we can predict a density of approximately 2.77 $g/cm^3$ (Figure 11).

Since our geophysical estimates are for the bulk crust, we also assume a reference crust, in which we consider only the top 50% (i.e., 15 km) of upper crust with a variable silica content. The rest of the crust is fixed, and it is composed by 25% (i.e. 7.5 km) of middle crust (intermediate composition) and the bottom 25% of lower crust (mafic composition). With this reference crust, we obviously get even higher values for the $V_P/V_S$ ratio, and we infer a silica content of the upper crust
ranging from ∼67% to 75% (we document this result with Figure S18, in supporting material). Moreover, the average crustal density becomes larger and reach a value around 2.8 $g/cm^3$ (Figure S18).

The silica-rich composition we infer for Variscan batholith is consistent with the chemical composition estimated for a portion of exhumed Variscan crust in Calabria (Sassi, 2003; Fiannacca et al., 2017). In addition, the Moho depth observed in the batholith correspond to that of the intact Variscan European crust. Our findings suggest that the subduction-driven processes
associated with the formation of the islands in the past 35 million years did not significantly modify the continental crust.

On the other hand, the western portion of Sardinia has a thinner crust and higher $V_P/V_S$ ratios, which are consistent with the presence of Cenozoic volcanic rocks. A remarkable finding is the identification of a $V_P/V_S$ ratio exceeding 1.9 in a region where an independent study reported the lowest $V_S$ values and where heat flow is significantly high. We propose that this region is characterized by elevated temperatures within the crust, potentially approaching the solidus.

The Corsica stations yield other important results. Stations located on the Alpine front in Corsica image a deep Moho, in agreement with previous studies, and high $V_P/V_S$ ratios. However, the $V_S$ values in these regions are not low. We interpret these findings as indicative of the presence of serpentinite-rich rocks throughout the crust, highlighting the significant influence of Alpine orogenesis processes on the entire crust of Corsica. The MORSI stations located in the extreme north of Alpine Corsica also has a high $V_P/V_S$ ratio, but a shallower Moho depth. This outcome is consistent with the physical properties of the
metamorphic Alpine rocks and with the partial sampling of the oceanic crust surrounding the station. Interestingly, the nearby CELB station on Elba island, within the Tuscany magmatic province, displays a shallow crust, as expected. However, it shows a lower $V_P/V_S$ ratio, which is consistent with the abundance of granodioritic and monzogranitic rocks forming a large pluton that intruded approximately 7 million years ago (Caggianelli et al., 2014).

## 5 Conclusions

Subduction-related processes play a fundamental role in plate tectonics. The rapid retreat of subducting slabs lead to the formation of volcanic arcs and the opening of back-arc basins. In certain cases, such as the example of Sardinia and Corsica, the process of rifting can cause segments of continental crust to separate from the main continent, becoming part of the newly formed basin. Our study reveals that these processes, which are responsible for extensive arc Cenozoic volcanism in Sardinia



and significant intraplate volcanism in the Tyrrhenian Sea, did not result in substantial modifications to the Variscan crust,
which forms a substantial portion of the Sardinia-Corsica microcontinent.

In contrast, the previous Alpine orogenesis, as evidenced by observations from two stations in Central Corsica located along the Alpine front, had a profound impact on the entire crust. This orogenic event, associated with the collision between the African and European plates, caused significant changes, including deformation and metamorphism, throughout the crust.

Furthermore, we have discovered compelling evidence of a thermal anomaly within the crust in the eastern part of Sardinia.
We find this anomaly in correspondence of a region characterized by several other elements that support our finding; namely a region that experienced relatively recent (1.5 m.y. ago) volcanism, current rifting, high heat-flow and low shear-wave velocity.

*Code and data availability.* The seismological data used in this work are partly publicly available through the European Integrated Data Archive (EIDA; http://www.orfeus-eu.org/data/eida/; EIDA, 2022) and partly come from in-house data of the LiSard seismic network. Full three-component waveforms and computed receiver-functions waveforms used in this study, as well as, scripts to reproduce the results are
available from https://www.dropbox.com/scl/fo/1nzvtybvbawnqs6hjpxdz/h?rlkey=1yz23d4u2cl827i8r0ncusnv0&dl=0 . In the same repository we also provide thermodynamic tables of physical properties computed with Perple-X and related scripts.

*Author contributions.* FC, IF, and MVM installed the stations and acquired the data; FC, AC, and HR computed and interpreted the receiver functions; FC performed the thermodynamics computations; FC wrote the the manuscript draft; HR, IF, MVM, AC and FC reviewed and edited the manuscript.

*Competing interests.* There are no competing interests

*Acknowledgements.* FC and HR acknowledge the Grant of Excellence Departments, MIUR-Italy (ARTICOLO 1, COMMI 314 - 337 LEGGE 232/2016).



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





**Figure 1.** Geological map (left panel), modified from Malusà et al. (2016), and topography map with the locations of seismic stations used in this study (right panel).





**Figure 2.** Radial and transverse P-wave receiver functions for two LiSard stations. Panels at the right show the backazimuth of the recorded events. In top panel, UT.003, whose sampling is similar to the other LiSard stations. In the bottom, the poorly sampled UT.011.





**Figure 3.** Radial and transverse P-wave receiver functions for two public stations. Panels at the right show the backazimuth of the recorded events. In top panel, MN.VSL, probably the most extensively studied station. In bottom panel, IV.AGLI, which shows noisy and/or heterogenous receiver functions.







**Figure 4.** Results of the H-k stacking process for 011, 009, 005 and 006 Lisard stations, all showing intricate features (see main text for details). Stack values in each panel are normalized from 0 to 1.





**Figure 5.** Results of the H-k stacking process for 6 high-quality Lisard stations. Stack values in each panel are normalized from 0 to 1.





**Figure 6.** Results of the H-k stacking process for 4 public stations in Corsica of the French (RESIF) network.







**Figure 7.** Results of the H-k stacking process for 5 public stations in Sardinia and 1 in Elba Island of the Italian (INGV) and the Mediterranean (MedNet) networks.



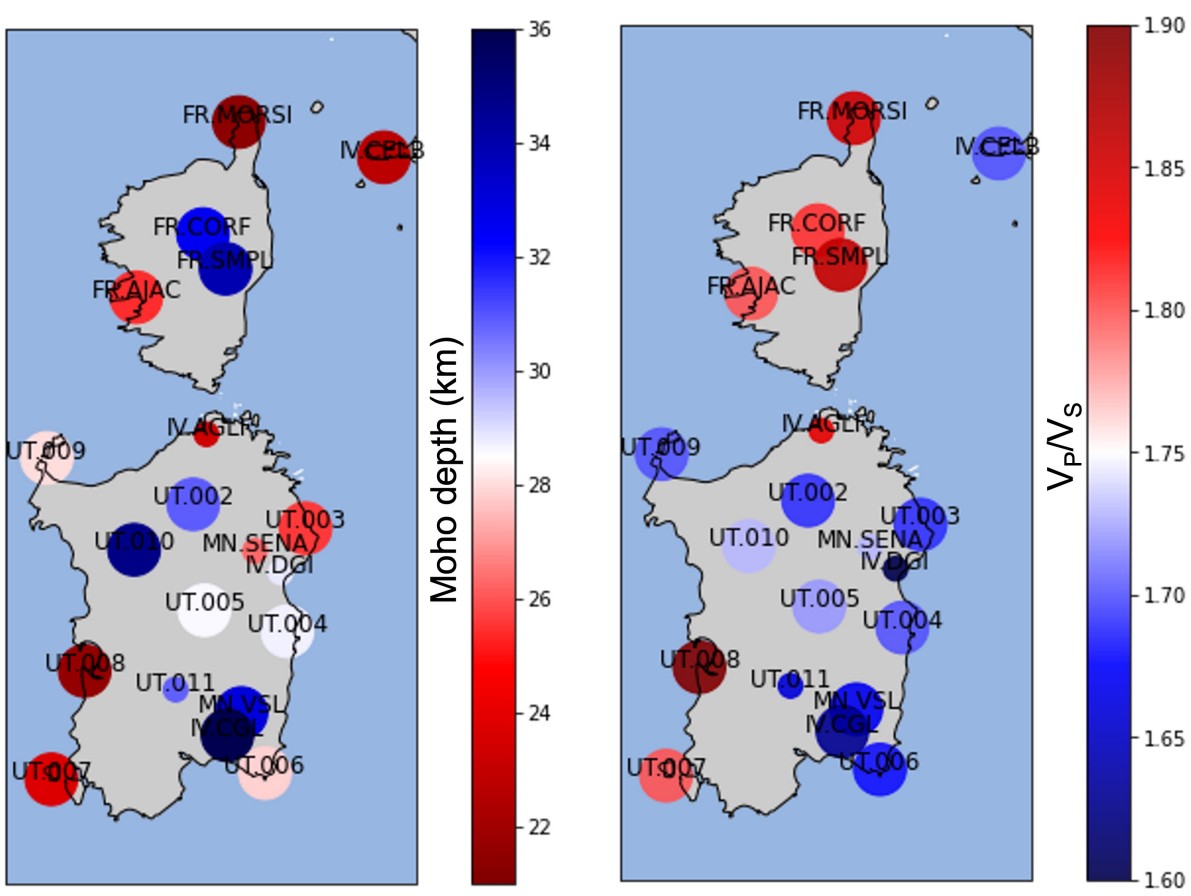

**Figure 8.** Moho depth (left panel) and $V_P/V_S$ ratio (right panel) obtained with the H-k stacking. Four most problematic stations are indicated with smaller symbols.

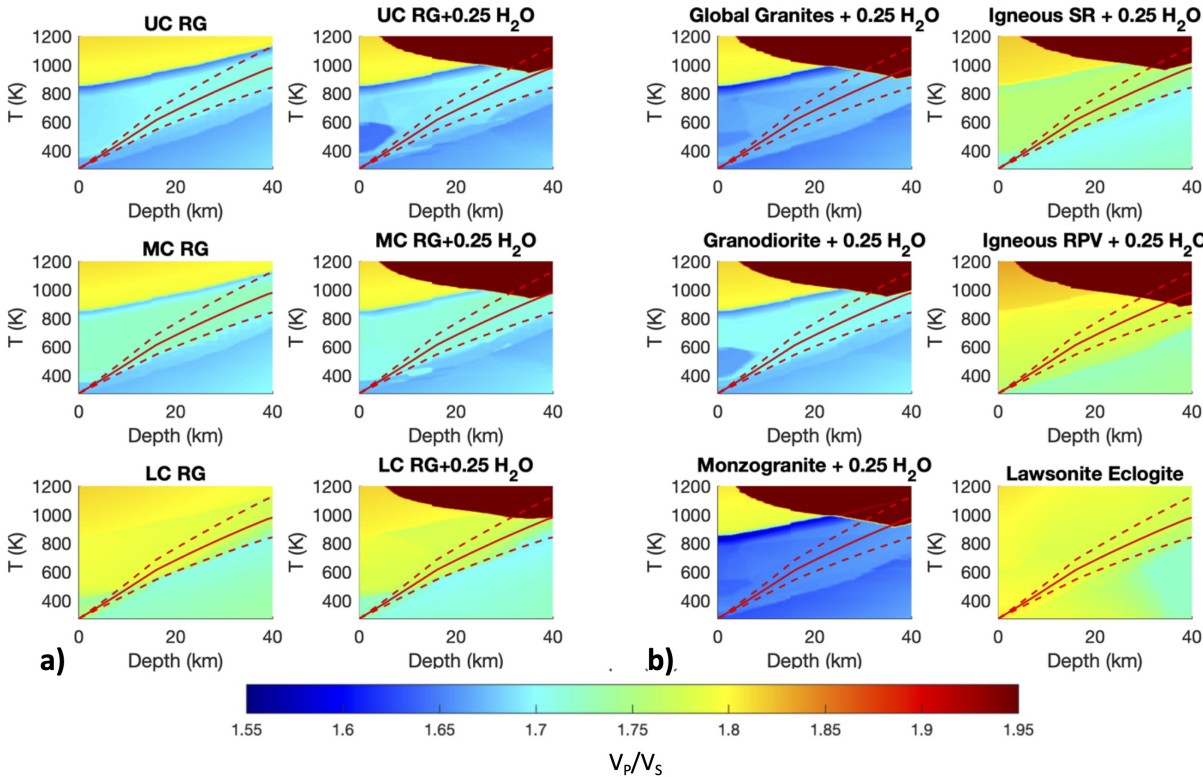

**Figure 9.** $V_P/V_S$ ratio as function of depth and temperature (T) for different chemical compositions. In each panel, we show reference thermal structures: solid red line is a continental geotherm assuming a $60\ mW/m^2$ heat flow, dashed lines are 40 and 80 $mW/m^2$ geotherms. Panels on left side (a) show predicted values for average upper-, middle- and lower-crust compositions from Rudnick and Gao (2014). Right side of panel (a) are the same composition with 0.25 wt% of $H_2O$. Panels b represent, on the left, estimated compositions of global granites (Blatt and Tracy, 1996) and for two most abundant rocks in the Sardinia-Corsica granitoid batholite, i.e. Granodiorite and Monzogranite (from Poli et al. 1989). On the right, we show values for chemical compositions representative of the igneous Cenozoic rocks outcropping in Western Sardinia (from Lustrino et al., 2013) and for a typical Eclogite (dry) of Alpine Corsica. All chemical compositions here considered are reported in Table 1.



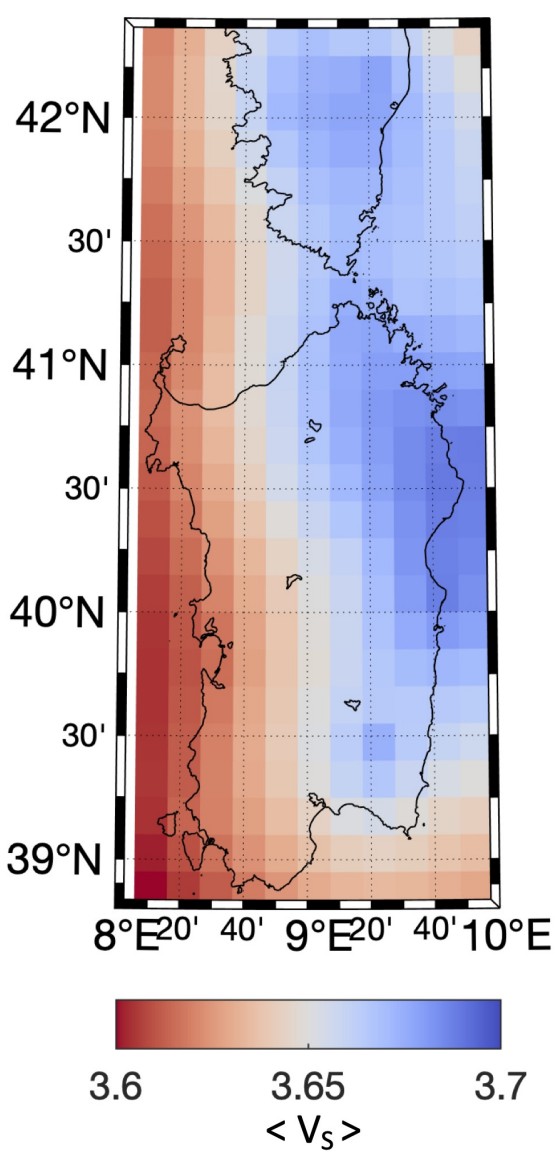

**Figure 10.** Average shear-wave velocity ($V_S$) until a depth of 30 km inferred from the $V_S$ model of Magrini et al. (2019).



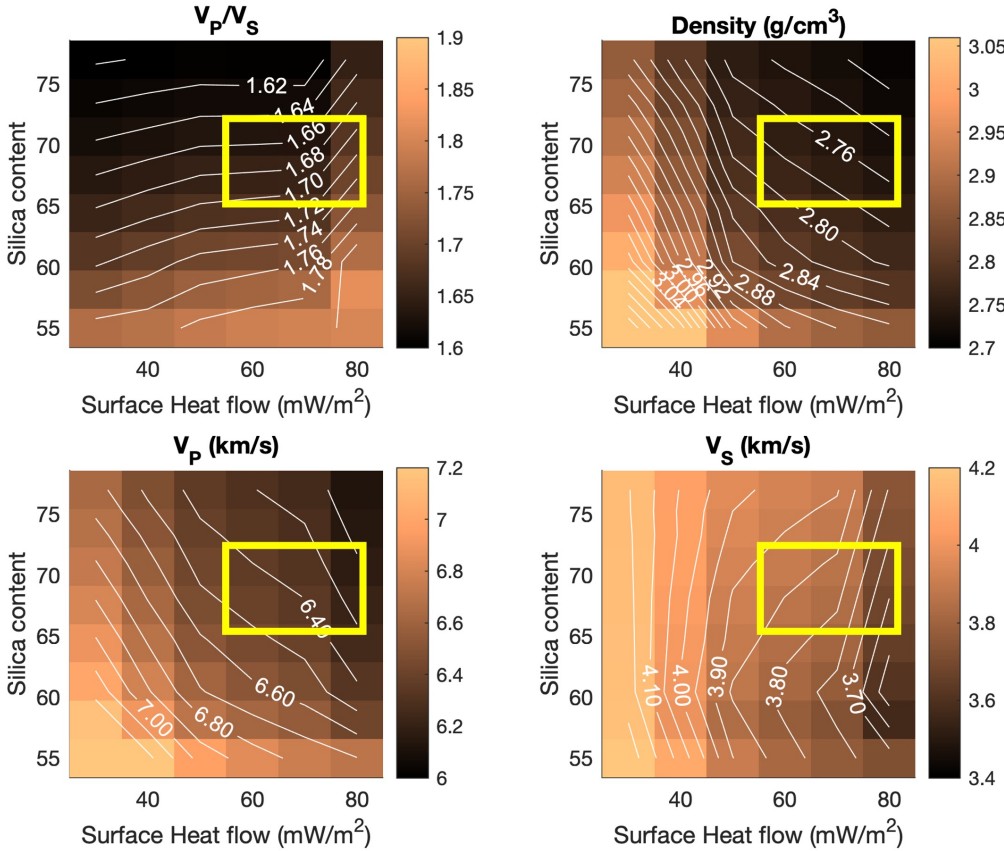

**Figure 11.** Effect of temperature and silica content on average $V_P/V_S$, absolute velocities and density of the crust. The values refer to the chemical compositions in Table 2, and have been calculated for a crustal depth of 30 km and for thermal structures based on continental geotherms at variable heat-flow. In yellow, we indicate the range which satisfy the physical properties as constrained by geophysical studies. In supporting material, we provide two similar figures, but with a variable silica-content upper-crust only in the top 15km.



**Table 1.** Chemical compositions (wt%) representative of different petrological provinces in the study area: Variscan batholith, Alpine Corsica and Cenozoic Volcanism of West Sardinia.

|  | $SiO_2$ | $Al_2O_3$ | FeO | $MgO$ | $CaO$ | $Na_2O$ | $K_2O$ |
|---|---|---|---|---|---|---|---|
| SG | 66.28 | 15.66 | 4.57 | 2.61 | 4.20 | 2.80 | 2.63 |
| SM | 74.59 | 13.93 | 1.51 | 0.51 | 1.76 | 3.22 | 4.22 |
| GG | 72.16 | 14.44 | 2.78 | 0.71 | 1.82 | 3.70 | 4.13 |
| SRV | 57.88 | 16.96 | 7.35 | 4.93 | 7.86 | 2.65 | 2.11 |
| RPV | 62.16 | 17.31 | 5.46 | 1.56 | 3.34 | 5.22 | 4.70 |
| LE | 46.10 | 17.70 | 6.46 | 3.73 | 14.35 | 3.15 | 0.15 |

All chemical compositions have been re-normalized to our 7-oxide system and have 0.25 wt% of $H_2O$. $FeO$ is total. SG: Sardinia granodiorites (Poli et al. 1989), SM: Sardinia monzogranites (Poli et al. 1989), GG: Global granites (Blatt and Tracy 1996), SRV: representative composition of Subduction-Related Volcanism (Lustrino et al. 2013 and references therein), RPV: representative composition of Radiogenic-Pb-Volcanism (Lustrino et al. 2013 and references therein), LE: Lawsonite Bearing Eclogite (Vitale Brovarone et al., 2011), outcropping in Alpine Corsica



**Table 2.** Chemical compositions (wt%) obtained by varying the silica content from the global composition of Rudnick and Gao for the upper crust.

|      | $SiO_2$ | $Al_2O_3$ | FeO  | $MgO$ | $CaO$ | $Na_2O$ | $K_2O$ |
|------|---------|-----------|------|-------|-------|---------|--------|
| RG a | 76.96   | 10.77     | 3.53 | 1.74  | 2.51  | 2.29    | 1.96   |
| RG b | 75.24   | 11.59     | 3.79 | 1.87  | 2.70  | 2.46    | 2.11   |
| RG c | 73.24   | 12.53     | 4.10 | 2.02  | 2.92  | 2.66    | 2.28   |
| RG d | 70.88   | 13.64     | 4.47 | 2.20  | 3.18  | 2.90    | 2.48   |
| RG e | 68.07   | 14.98     | 4.90 | 2.41  | 3.49  | 3.18    | 2.72   |
| RG f | 64.65   | 16.59     | 5.43 | 2.67  | 3.87  | 3.52    | 3.02   |
| RG g | 60.40   | 18.60     | 6.09 | 3.00  | 4.33  | 3.95    | 3.27   |

All chemical compositions have been re-normalized to our 7-oxide system and have 0.25 wt% of $H_2O$. $FeO$ is total. Composition goes from $\sim$ 60 to 76 wt% of $SiO_2$. Physical properties as function of $SiO_2$ and thermal gradients are shown in Figure 11.





**Table 3.** Number of total events analyzed for each location and final number of receiver functions which passed our selection criteria and visual inspection (see main text). In total, 6959 P-wave receiver functions have been considered. 2573 passed our selection criteria for further processing. In gray, the UT.001 station, that has been discarded for further processing.

| Station | Total events | Selected after quality control | Selected after visual inspection |
| --- | --- | --- | --- |
| *UT.001 (LaMaddalena)* | 132 | 35(11) | 11 |
| UT.002 (Oschiri) | 221 | 94(75) | 69 |
| UT.003 (Siniscola) | 208 | 126(112) | 108 |
| UT.004 (Lotzorai) | 206 | 118(99) | 97 |
| UT.005 (Ovodda) | 213 | 92(76) | 69 |
| UT.006 (Villasimus) | 204 | 104(94) | 90 |
| UT.007 (Carloforte) | 223 | 116(95) | 89 |
| UT.008 (Capofrasca) | 224 | 112(89) | 83 |
| UT.009 (Asinara) | 232 | 111(97) | 93 |
| UT.010 (Giave) | 227 | 136(98) | 91 |
| UT.011 (Villafranca) | 87 | 38(16) | 14 |
| FR.MORSI | 727 | 438(333) | 316 |
| FR.AJAC | 793 | 513(365) | 320 |
| FR.CORF | 163 | 103(74) | 66 |
| FR.SMPL | 798 | 511(389) | 370 |
| IV.CELB | 318 | 207(153) | 150 |
| IV.AGLI | 623 | 361(44) | 32 |
| IV.DGI | 384 | 251(79) | 70 |
| IV.CGL | 195 | 128(74) | 62 |
| MN.VSL | 616 | 424(314) | 299 |
| MN.SENA | 165 | 104(79) | 74 |



**Table 4.** Inferred Moho depths and $V_P/V_S$ ratio and their confidence intervals from the H-k stacking for all the stations. We use a fixed $V_P$=6.4 km/s and we assign a weight equal to 0.6 ,0.3 and 0.1 for the Ps, PpPs and the PpSs+PsP phases, respectively, for all the stations except three. UT.005 and UT.006 are weighted differently (i.e, 0.8, 0.1, 0.1). For Station IV.DGI we only use the Ps phase. We also add an estimate of average shear-wave velocity ($<V_S>$) and variations according to Moho uncertainties from the model of Magrini et al. 2019 and the Moho depth obtained by assuming a fixed $V_P/V_S$ ratio equal to 1.7.

| Station | Lat. | Lon. | Height (m) | Moho (km) | CI Moho (km) | $V_P/V_S$ | CI $V_P/V_S$ | $<V_S>$ (km/s) | CI $<V_S>$ (km/s) | Moho (km) at $V_P/V_S$=1.7 |
|---|---|---|---|---|---|---|---|---|---|---|
| UT.002 | 40.717 | 9.096 | 193.0 | 30.9 | 1.8 | 1.69 | 0.07 | 3.69 | 0.030 | 30.7 |
| UT.003 | 40.577 | 9.753 | 110.0 | 25.6 | 1.8 | 1.69 | 0.08 | 3.61 | 0.032 | 25.4 |
| UT.004 | 39.974 | 9.647 | 725.0 | 28.7 | 1.7 | 1.70 | 0.07 | 3.66 | 0.029 | 28.7 |
| UT.005 | 40.097 | 9.161 | 69.0 | 28.6 | 5.3 | 1.72 | 0.17 | 3.64 | 0.087 | 29.2 |
| UT.006 | 39.144 | 9.516 | 48.0 | 27.8 | 4.7 | 1.68 | 0.13 | 3.61 | 0.089 | 27.3 |
| UT.007 | 39.108 | 8.266 | 55.0 | 23.7 | 1.8 | 1.80 | 0.09 | 3.50 | 0.035 | 25.4 |
| UT.008 | 39.742 | 8.461 | 93.0 | 21.6 | 2.5 | 1.91 | 0.11 | 3.46 | 0.053 | 27.0 |
| UT.009 | 40.988 | 8.240 | 5.0 | 28.0 | 2.5 | 1.70 | 0.15 | 3.59 | 0.042 | 28.0 |
| UT.010 | 40.451 | 8.750 | 622.0 | 34.8 | 1.4 | 1.73 | 0.06 | 3.72 | 0.021 | 35.1 |
| UT.011 | 39.633 | 8.993 | 295.0 | 30.8 | 2.1 | 1.66 | 0.12 | 3.66 | 0.033 | 30.3 |
| FR.AJAC | 41.928 | 8.763 | 27.0 | 25.4 | 1.1 | 1.80 | 0.05 | 3.58 | 0.022 | 29.5 |
| FR.CORF | 42.298 | 9.153 | 475.0 | 32.5 | 1.7 | 1.81 | 0.08 | 3.71 | 0.034 | 37.5 |
| FR.SMPL | 42.095 | 9.285 | 405.0 | 33.9 | 1.0 | 1.86 | 0.04 | 3.75 | 0.020 | 41.0 |
| FR.MORSI | 42.953 | 9.363 | 111.0 | 21.4 | 0.8 | 1.85 | 0.06 | — | — | 25.7 |
| MN.SENA | 40.444 | 9.457 | 338.0 | 26.3 | 1.3 | 1.73 | 0.07 | 3.62 | 0.023 | 28.6 |
| MN.VSL | 39.496 | 9.378 | 370.0 | 33.0 | 1.3 | 1.67 | 0.04 | 3.72 | 0.022 | 32.5 |
| IV.CELB | 42.747 | 10.211 | 742.0 | 22.7 | 1.1 | 1.70 | 0.06 | — | — | 22.7 |
| IV.AGLI | 41.127 | 9.173 | 180.0 | 23.4 | 7.5 | 1.84 | 0.22 | 3.56 | 0.127 | 29.0 |
| IV.DGI | 40.318 | 9.607 | 354.0 | 28.8 | 7.0 | 1.58 | 0.20 | 3.66 | 0.124 | 25.0 |
| IV.CGL | 39.366 | 9.296 | 1050.0 | 35.9 | 1.6 | 1.63 | 0.07 | 3.77 | 0.027 | 32.5 |