# Peer review of "On crustal composition of the Sardinia-Corsica continental block inferred from receiver functions"

_EGUsphere, 2024_

## Author Comment (AC1)

Dear Editor, please find enclosed a revised version of our paper, in which we address all the reviewer comments. We have attached a version of the paper with modifications highlighted in red. Most of the figures have been improved compared to the previous version. Additionally, we have corrected an error in Table 2 regarding the modified compositions of the upper crust with varying silica content.

**Reviews Solid Earth**

We would like to thank Prof. Tony Lowry and an anonymous reviewer for their constructive comments, which helped us to improve the paper. Below, we provide detailed answers to each point and indicate how the revised version of the paper has been modified accordingly.

**First review (anonymous):**

Review of the manuscript "On crustal composition of the Sardinia-Corsica continental block inferred from receiver functions" by Fabio Cammarano, Henrique Berger Roisenberg, Alessio Conclave, Islam Fadel, and Mark van der Meijde submitted for publications to Solid Earth.

This work presents a newest addition to the investigation of the complex crustal structure of Sardinia-Corsica continental block using the P receiver function method at 21 seismic stations. For this the authors use data from both permanent stations and temporary ones from the project LiSard.

The topic of investigation is interesting, and the methods used are sound and my main critique of the manuscript is mostly about the presentation of the results and the usage of the receiver function method. The language used is appropriate and reads easily but the overall impression is that of the manuscript lacks a bit more depth in the analysis of the results. The authors compare the receiver function modeling results with crustal composition obtained by thermodynamical modeling but a bit more discussion on various influences of complex structure on receiver functions would make the main conclusions more robust.

Bellow, please find the comments that I suggest authors address before the manuscript can be considered for publication.

 Comments:

-Introductory section is well written with abundant information about tectonics and previous investigations in the area, but, in my opinion, accompanying Figure 1. is rather poor with part of the text in both images a) and b) not readable. Also, image showing seismic stations used should be plotted in more presentable way. Ttry some of the plotting packages like GMT,

Matplotlib basemap or variety of similar packages. Also, please put in the Figure caption what are red and what are black station markings in the station plot.

The authors should invest a bit of time to fix this as good introductory image will help reader connect conclusions with research aims.

We have modified the right panel of Fig. 1, which now shows a high-resolution topography and bathymetry map with locations of seismic stations, color-coded by network to highlight our in-house LiSard stations and publicly available stations used in this study. We used pygmt to produce this map (the python script is provided for reproducibility). We did not modify the geological map, which is based (and modified) on Malusà et al. 2016

Line 54: change "The average P-wave velocity…" to "The average crustal P-wave velocity…"

Done

- Data and methods section

Lines 99-100: In this sentence It is a bit unclear what is deconvolved from what. Please make it more concise.

There was indeed a typo, and the sentence was unclear. We have revised it.

Table 1: Why is the part of the caption above and part of it bellow the table? I suggest authors correct this and put everything above the table.

Done

Also, it is not clear which of the six groups is part of the 3 provinces mentioned in the caption?

We clarified in the caption which compositions belong to the Variscan batholith, Alpine Corsica and Cenozoic Volcanism of West Sardinia.

Line 178: Links to the software packages should go either in the Code and the data availability part after the main text or in the References section.

We add the links to the data availability section.

Table 3 should be moved to Supporting material as it will clutter the main body of the text without bringing any substantially relevant information that is not already shown in Figure 1.

We believe that Table 3 provides useful information, as it shows the total and selected number of waveforms after two quality-control criteria, giving readers insight into the relative "quality" of the inferred receiver functions.  For this reason, we chose to keep it in the main text.

Line 209: "…as indicated in Table 1." The authors probably mean Table 3 not Table 1.

Yes, we thank the reviewer for spotting this error, that we have now corrected.

Line 212: Correct reference from Zhu and H (2000) to Zhu and Kanamori (2000) and do the same in the Reference section.

 Done

Line 215: Please provide reference for the statement in this sentence.

We think that the reviewer refers to the statement: "... a variation of 0.1 in the latter causes about 4km change in the crustal thickness."  This trade-off was estimated by us, so we have clarified this in the revised version of the paper.

Figure 2 and Figure 3 (and figures S1 – S17) These figures should be made more presentable as currently some of the important details are not visible. The amplitudes of the main Ps arrival and reverberation are not visible at all. I suggest that authors make figures bigger by removing unnecessary text on Y-axis (listing of event) and remove last image showing Back-azimuth for each event and sort both R and T receiver functions by Back-azimuth. Why is currently only T-component sorted by back-az and R not?

In this way figure will be bigger and details clearer.

In the previous version, we used the default SeisPy visualization, which we found generally clear. However, we agree that this format was not optimal for all stations. We have thus followed the reviewer's suggestions to improve the figures. We enlarged the fonts, we removed event names and we amplified the waveforms. The quality of the figures has also been improved, as in the previous version we realized that they were blurred.

P.S. Note that both the R and T components were already sorted by back azimuth. In the R plot, it was listed the event name and in the T plot the back-azimuth for that event.

- Results section

Lines 234 – 235: "...would be only made SV (S-wave in the vertical plane) energy."  Wording here is a bit unclear. Please make it more concise and clearer.

We have rephrased this sentence

Figure 2-3 (and S1 – S17) Authors show T-component RFs but are not actively discussing them. TRFs are crucial in interpreting possibly 3D structural complexities (dip, anisotropy, etc.) especially important in H-Kappa stacking and interpretation of these results. Authors should spend some time to at least try to interpret some of the signal on T-component for stations that do not have clear maximum in H-k stacking (e.g. station IV.DGI or IV.AGLI).

We thank the reviewer for the suggestion. We recognize the importance of transverse RFs for understanding dipping structures and anisotropy, and we agree this analysis is highly

valuable. However, as we noted in the manuscript, this analysis is the subject of future (and ongoing) work. We have invested time in understanding why H-K stacking proved complex for some stations, often due to heterogeneity, dipping structures, and anisotropy. To disentangle the anisotropic component, we conducted a harmonic decomposition. However, this analysis indicated not-significant variations, particularly in the Vp/Vs ratio, consistent with expected results (see, for example, results from the H-k-c stacking by Li et al., 2019, JGR). Additionally, harmonic decomposition was only feasible when a statistically significant number of high-quality waveforms were available. While analyzing the transverse component is not in our goals, we have added a sentence on our observations from H-K stacking when attempting to isolate anisotropy.

Overall, discussion about possible influences on interpretation of RFs H-k stacking is thin and should be done more thoroughly as conclusions are based on these results that could be overinterpreted.

We agree that H-k stacking is prone to errors and that a thorough analysis could help. In any case, we are confident that the method is effective in this case. The VP/VS ratio obtained in Sardinia and Corsica align reasonably well with petrological expectations, and the Moho depth is consistent with finding from independent studies. Importantly, the observed Vp/Vs crustal dichotomy in Sardinia provides new insights into crustal composition within this continental microplate. In response to the reviewer's point, we have slightly extended the discussion regarding the reliability of our H-K stacks.

Lines 241 – 242 "...those show a noisy pattern." How is this estimated? How do the authors see that something is noisy and other stations are not? This needs to be more concise.

We have rephrased and explained what we meant by "noisy pattern". We agree with the reviewer that the term 'noisy' was an awkward choice in this context.

Figure 4-5-6-7:

In my opinion there are too many figures for showing H-k stacking. It would be better if the authors show only 4 relevant H-k stackings for stations and the rest can be moved to supporting information. Additionally, if there is a discussion about some the problems at particular stations with H-k that (or those) stations can be showed in separate Figure indicating possibly problems.

Also, text in the Figures and on both axes is small and hard to read.

We have partially implemented the reviewer's suggestion. The original figures have been moved to supporting information, and only the H-K stacked panels remain in the main text. We have also enlarged the font size and labeled each panel with the station name.

Line 260:

Authors are using station UT.011 in discussion and deeming that station as showing intricate features as on the other hand they dismissed that station on Line 241. as noisy? Why use it if it is noisy?

As stated earlier, we have remove the term "noisy" to explain complicated features in the receiver functions. We now state: "The only exception being UT.011 (Figure 2, bottom panel), which has a limited number of records satisfying the quality criteria adopted, and even those show some inconsistency. For instance, reverberations are not always at the same distance and amplitudes on the transverse RF are high."

Line 281:

"...for station UT.009, the search had to be extended to k = 1.5." Why it had to be extended to 1.5 when the maximum is at 1.7? There are several such inconsistencies through the manuscript connected with h-k that needs to be dealt with. Why hasn't all been calculated in the same broad range?

As shown in Fig. 4, for UT.009, the 95% confidence interval extends to values below 1.55, necessitating an adjustment in the parameter search. We reviewed each choice in the H-K search and ensured all were justified and based on specific criteria. We're unsure which specific inconsistencies the reviewer references, but we checked to ensure each choice is well-supported.

Lines 289-290: "...deeper Moho depth." wording depth is redundant. Put "...deeper Moho."

 Done

Lines 290 – 291: remove "indicating a substantial Moho depth for both locations." as it is already stated in the previous sentence.

 Done

Lines 294 – 295:

"Additionally, it is worth noting that the CORF station exhibits an indistinct Ps phase, implying the presence of intricate wave propagation characteristics specific to the Central Corsica region."

Or more likely complex structure.

If we understand well, the reviewer is referring to the fact that the complex structure can be related to observed receiver functions. If this is the case, we think that the reviewer's point is

well taken. The complex wavefield is indeed likely due to structural complexity, which we discuss later in the manuscript.

Line 307: "...situated at a depth of 111 m..." borehole or underground cave station? Give a couple of words to describe it.

We explicitly say now that the station is located inside a dismissed mine gallery.

Lines 307 - 311: Authors indicate possible problems with this station that may be connected with orientation problems. Please check.

There are no current problems related to orientation with the MN.SENA. A problem with the instrument response was identified by us during the first year of installation (2019) when the station was named IV.SENA. We had a confirmation of this problem with personal communication with personnel of INGV (Dr. Marco Olivieri). The present study uses data from MN.SENA, which started recording in 2021 and does not have these issues.

Lines 312 - 317: In the case of station IV.DGI if only Ps phase was used for H-k stack then the authors should not use the resulting Vp/Vs values from that stacking as a viable results as the uncertainty is too great.

If multiples are not used, there is a complete trade-off between Moho depth and crustal VP/VS, as noted in the text and shown in Fig. 7 (DGI panel). As reported in Table 4, we estimated a broad confidence interval for this station, with VP/VS ranging from 1.38 to 1.78.

Line 341: "Additionally, we have computed, the P-T properties of the upper-, middle- and lower-crust from the global compilation of Rudnick and Gao (Table 2)."

The sentence is a bit misleading as one expects that the authors show P-T properties in Table 2. Correct this please.

We have rephrased it to indicate that Table 2 reports the global compilation of Rudnick and Gao

Line 376: "...average Vs in the eastern part." Figure 10 shows that western side has lower Vs values?

We thank the reviewer for catching this error. Yes, Vs is higher (and not lower).

Line 383: "...slightly from 3.75 to 3.80..." In Figure 11. one can see that these values are approx. 3.85 to 3.90.

Thank you for identifying this discrepancy. We have corrected the values to 3.82 to 3.87.

---

## Author Comment (AC2)

Dear Editor, please find enclosed a revised version of our paper, in which we address all the reviewer comments. We have attached a version of the paper with modifications highlighted in red. Most of the figures have been improved compared to the previous version. Additionally, we have corrected an error in Table 2 regarding the modified compositions of the upper crust with varying silica content.

**Reviews Solid Earth**

We would like to thank Prof. Tony Lowry and an anonymous reviewer for their constructive comments, which helped us to improve the paper. Below, we provide detailed answers to each point and indicate how the revised version of the paper has been modified accordingly.

**Second Review (Tony Lowry)**

This paper is a well-written and interesting analysis of crustal Vp/Vs from receiver functions in the Corsica-Sardinia microplate, with potential implications for processes and timescales of metasomatic alteration of continental crust. The authors find low Vp/Vs < 1.75 in the Variscan batholith of most of Sardinia where crust is thicker, and higher Vp/Vs in Corsica and the southwesternmost part of Sardinia. These patterns are interpreted in terms of silica content and mafic to ultramafic, supported by thermodynamical modeling of properties, but perhaps a more fruitful way to interpret these (as detailed further below) is in terms of quartz abundance reflecting a history of metasomatism in low Vp/Vs regions and more mafic lithologies elsewhere.

We thank Prof. Lowry for this suggestion. We agree that quartz abundance is a good alternative for parametrizing the compositional effects within the crust.

There are pros and cons in using a thermodynamical modeling approach. Our 'thermodynamically equilibrated' mineralogies show good sensitivity to Vp/Vs ratios. However, as discussed in our previous works (Guerri et al. 2015, Diaferia et al. 2017), assuming thermodynamic equilibrium within the crust, particularly in the upper crust, is often a strong assumption. That said, thermodynamic modeling allows us to incorporate pressure and temperature effects on both the elastic properties of minerals and phase equilibria - something most empirical relationships fail to address, as they are generally based on measurements at ambient temperatures and moderate pressures. Furthermore, thermodynamic modeling can provide valuable insights into the average chemical composition of the crust (in this case for the Sardinia -Corsica microplate).

While alternative parametrizations are possible, our approach here was varying the silica content in the upper crustal composition to investigate its effects on Vp/Vs ratios (with the aim to estimate the silica content of the entire crust). The Vp/Vs ratio is indeed due to the variable presence of quartz between the tested composition. We also agree that quartz can

be more abundant in regions where metasomatism was strong. Testing the effect by varying water content is also possible, however, from a perspective of physical properties, we note a slight change in quartz abundance if we vary the water content.

In the appendix at the end of this response letter, we include plots showing the volume percentage of alpha quartz as a function of pressure and temperature for compositions with varying silica contents and two different water contents.

One caveat worth making in the paper, and remembering in the context of interpretation, is that uncertainties in single-station estimates of Vp/Vs using H-k stacking techniques can be quite large. For example, raw one-sigma uncertainties are greater than 0.1 based on the variance at ~0 distance separation for variograms of USArray data in the automated EARS database (Crotwell & Owens, SRL 2005), see e.g. Fig. 2c of Lowry & Pérez-Gussinyé (Nature 2011). The authors appear to have done a very careful analysis here, and judging by the stacks in Figures 4-7 the younger crust of the study area is less structurally complicated than typical North American continental crust in USArray, but even so uncertainties are likely to be of order 0.07 or larger here and possible impacts of that should be addressed in the discussion.

We agree with the reviewer that H-k stacking can be problematic in regions with complex structural variations. However, in our study, despite the limited number of receiver functions available, we were able to obtain robust results for most of them. As the reviewer noted, we made every effort to carefully analyze the waveforms, though we acknowledge that uncertainties might still affect the interpretation. Consequently, we have added a couple of sentences regarding the inherent uncertainties in the Vp/Vs estimates.

Our interpretation, however, integrates previous and independent seismological results. In particular, we include constraints on average Vs derived from surface-waves dispersion curves based on ambient noise, as well as heat-flow and petrological constraints.

There are a couple of other issues that it might be worthwhile for the authors to consider in a revision of the paper, described in greater detail in comments tied to §2.1.1 below. One is that there appears to be some sort of bias error in Perple_X outputs of Vp/Vs for crustal mineral assemblages, resulting in much lower modeled values than those measured in the lab for corresponding rocks. The practical significance of this is that an exotic (serpentinite, eclogite, or supersolidus) lithology is not necessary to explain higher Vp/Vs in the study area; a gabbro would be sufficient. However it also means that the absolute values of Perple_X-derived Vp/Vs are less useful for interpreting these results than how Vp/Vs changes for different chemistries and volatile contents. I would also suggest that it's useful to recognize that Vp/Vs variation is dominated not by %-Si so much as by %-qtz, because of the unique elastic properties of quartz. This enables the use of Vp/Vs as a proxy for metasomatic history, as much of the Vp/Vs difference for dry and hydrated lithologies in Figure 9 is related to breakdown of feldspar to quartz and mica (Ma & Lowry, 2017).

This is an interesting discussion point. The reason for the discrepancy between results of physical properties obtained with thermodynamic modeling and experiments is indeed not

resolved. From one side, shear properties of crustal minerals are not so well constrained experimentally, and probably some efforts from the experimental mineral physics community should be envisaged on this aspect. The database we used (details in Cammarano and Diaferia 2017), although appositely conceived for crustal physical properties, needs to be improved.

That said, the discrepancy between empirical experimental laws and thermodynamic calculations may be due to several factors, which is difficult to address systematically. For example, saturated cracks and anisotropy can increase the Vp/Vs ratio in the shallow portion of the crust (Wang et al. 2012).  Additionally, crustal VP/VS ratios often deviate from empirical relationship (such as the one from Brocher 2005), as shown in Figure 5 of Diaferia et al. 2019 - JGR) and this needs to be taken into account.

Despite the uncertainties in modeling, we were positively surprised by the good match between the seismically measured and modeled average crustal properties (see Fig. 9 and 11 of the paper). An exception was observed for two stations in Corsica. Here we identified a deep Moho, consistent with previous findings, but we found a much higher Vp/Vs ratio in this region (despite the outcropping granitoids very much alike those of the Sardinian Variscan basement). The inferred Vp/Vs value falls within the gabbro range, but we interpret it in the context of significant serpentinization of the Corsica crust, supported by both the high Vp/Vs ratio and the high average Vs found in a previous study, as well as the abundant serpentinite-rich rocks in Alpine-Corsica. Of course, uncertainties in the estimated VP/VS (see Table 4) are still large enough to make it difficult to reach a firm interpretation.

In response to the reviewer's last suggestion, in our parametrization, we focus on varying the total SiO2 content, which mostly affects the abundance of quartz. We have included in the appendix some results showing the abundance of alpha quartz as function of pressure and temperature for the compositions used in Figure 11. In these calculations, we fixed the water content to 0.25 wt%. While the addition of water is significant for several aspects, its main effect on seismic properties is in lowering the solidus, as we have documented in previous work (Guerri et al 2015, Diaferia et al. 2017 ). In the appendix, we also provide additional plots showing the quartz abundance for composition with higher (~1 wt%) water content.

**§ 2.1.1 Thermodynamical (Perple_X) modeling:**

The choice here to examine various bulk compositions but only use one constant (0.25wt-% H2O) volatile state unfortunately obscures one of the most significant potential takeaways for interpretation of the results. Namely, the primary factor in determining bulk crustal Vp/Vs is the abundance of the mineral quartz (Christensen, JGR 1996; Lowry & Pérez-Gussinyé, Nature, 2011). This does of course depend to some degree on SiO2 content, but it is much more sensitive to whether water is present to react with the bulk constituents, which breaks down feldspar into quartz and mica (Ma & Lowry, Tectonics, 2018). Hydration reactions that break down feldspar also presumably depend on whether CO2 is present to buffer those

reactions (Yardley, J. Geol. Soc. Lond. 2009). From that perspective, it seems to me that a more useful approach to examining Vp/Vs with Perple_X is to use the bulk compositions from the rock environment of interest but vary the volatile mix, and then interpret the variations primarily in terms of hydration history.

As mentioned earlier, we present some tests with a different water content in the appendix. We also discussed in the paper the possibility of modeling the composition in terms of hydration history. Of course, it is difficult to separate the two and I agree that hydration is also a factor. However, especially for the studied region, we believe it is more useful to provide an overall view of the chemical composition of the crust.

Also, as an aside: There is a problem of some sort in the elastic parameter database of Perple_X, because it gives Vp/Vs seismic velocity ratio estimates that are consistently about 0.05 to 0.1 lower than the corresponding values from Christensen's (JGR 1996) measurements. This becomes apparent if one compares the 1.71 to 1.86 range of Christensen's measurements of granite to gabbro in Fig. 1a of Ma & Lowry to predictions in Figure 9 of this paper. For that reason, Ma & Lowry did not show absolute Vp/Vs in their Fig. 14, but rather the perturbations with water versus without water present in Perple_X thermodynamical modeling that used a similar database to this paper. Xiaofei Ma spent significant time and effort trying to figure out where the problem may be coming from during his dissertation studies, but we were unable to track it down. Since attenuation effects are likely to be larger for Vs than Vp, that is one candidate for the discrepancy, but I am somewhat skeptical that attenuation would be that significant for small-scale room temperature samples like those in Christensen's (JGR 1996) database. Because of this, it is perhaps safer to use Perple_X as a tool to examine relative Vp/Vs for different choices of composition or state than for purposes that assign meaning to the absolute Vp/Vs. For purposes of this paper, the conclusions regarding likely mineral assemblages are probably still valid (within the large uncertainties that are inherent in single-station H-k stacking estimates of Vp/Vs), but note that robust measurements of whole-crustal averaged Vp/Vs less than ~1.7 are extremely rare except when errors are present due to the perturbations of amplitude stacks by other reflectivity, dipping structure and anisotropy, as this paper notes can be present. Vp/Vs exceeding 1.8 on the other hand does not require an unusual composition like that of eclogite; it simply requires lower abundance of quartz. In fact, the mean Vp/Vs for the USArray footprint in the United States is about 1.79 (Ma & Lowry, Tectonics 2017) and Vp/Vs exceeding 1.85 is possible in crust where lithologies are gabbroic.

We understand the importance of achieving consistency between experimental data on rocks and thermodynamic properties. However, as mentioned earlier, the variations are due to several factors, and I agree that tracking them down is challenging. The role of anelasticity in the crust is matter of debate. We have been working on evaluating the attenuation of Rayleigh waves in the crust, and our results for the US suggest that fluid-filled fractures play an important role in governing attenuation of the upper crust rather than temperature (see Magrini et al. 2021). This could potentially affect the Vp/Vs ratio (see Wang et al. 2012),

increasing values in tectonically active regions. This finding is consistent with our previous work from a qualitative perspective (Diaferia et al. 2019). While this is not a primary focus of the current paper, I agree that it is one of the factors to consider.

.

I would also like to express a more optimistic view than Prof. Lowry. In fact, I found the discrepancy between the values obtained with different methodologies to be smaller than expected.

Regarding the inferred high VP/VS values, we interpret them differently. Where we observed very high Vp/Vs values in Sardinia, we also observe a thin crust and, previously, we detected a very low Vs. In central Corsica, the situation is different: here we found high Vp/Vs values, but without anomalous Vs and with a thicker crust. This, combined with petrological and geodynamic constraints, supports our interpretation, although it is not yet conclusive.

We have now added a sentence to better explain why we suggest the serpentinization of the crust in Corsica may play an important role. **In addition, we recognize that a mostly mafic crustal composition might be sufficient.**

Lines 368-369: The change in Vp/Vs due to adding water to the chemistry is not primarily because of the reduction of the solidus, as inferred here, but because hydration reactions reduce the feldspar content of the mineral assemblage and instead favor formation of quartz and mica (Ma & Lowry, Tectonics 2017). The abundance of quartz dominates crustal Vp/Vs variations because quartz has a very unusual Poisson's ratio translating to a Vp/Vs less than 1.5 (Lowry & Pérez-Gussinyé, Nature 2011, based on measurements in Christensen, 1996).

We agree that this is a very important effect. As shown in Guerri et al.2015, the breakdown of plagiocase produces a visible jump in physical properties, which can even explain some crustal seismic transitions. The abundance of quartz is affected but not as much by water content (see added tests the appendix). In general, our parametrization in terms of silica content can explain observations providing useful infos on the chemical composition of the crust.

Lines 415-418: As noted above, a high Vp/Vs coupled with high Vs does not require serpentinite or some other exotic mineralogy to explain; rather a Vp/Vs up to 1.88 and high shear velocity can be expected for a common mafic lithology. Where Vp/Vs exceeds 1.88, it can probably be attributed to the large uncertainties expected for single-station H-k stacking estimates of Vp/Vs.

We still prefer our interpretation, not because we can dismiss the reviewer's suggestion, but because different independent datasets seem to support the presence of serpentine-rich crust

beneath Corsica. However, we acknowledge that, given the uncertainties in H-k stacking and modeled Vp/Vs, an alternative common mafic lithology cannot be ruled out.

**§ 5 Conclusions**

This section does not tie back strongly to the information presented in the previous sections. It seems to me that the modeling in the discussion section does reaffirm earlier work suggesting that hydration lowers Vp/Vs, and so– given that Vp/Vs in Corsica and southwestern Sardinia remains high– however-much hydration may contribute to regional volcanism, it does not appear to translate to substantial metasomatic modification of the crustal lithology in those particular locations (which is slightly different than what is said in lines 425-430). I also don't know that I would strongly emphasize evidence for a thermal anomaly in Sardinia, given that Vp/Vs is insensitive to temperature. (And although it may be sensitive to partial melt, in practice the crustal averages of Vp/Vs derived from H-k stacking are not.)

We recognize that there is an alternative explanation for the location in Sardinia where we observed high Vp/VS, high Vs, and thin crust. In this case, the potential role of sediments coupled with porosity and fluid-filled cracks, could explain the observations without invoking a thermal anomaly.  However, it is interesting to note that this portion of the island also shows high heat flux and has been interested by relatively recent volcanism (plio-pleistocenic).

Regarding the first point, we have now provided a more detailed explanation of our results.

Figures 2 and 3: These are a bit hard to interpret because the y-axis back-azimuth for the left and center panels is nonlinear, and must be inferred by looking back and forth to the right panel. It would make more sense to bin and sum, or if showing individual traces is preferred, to plot each trace as a linear function of back-azimuth (even if they then overlap). This would aid in identifying patterns expected for dipping layer boundaries and/or layer anisotropy (e.g., Schulte-Pelkum & Mahan, EPSL 2014).

We like to show all the original waveforms as function of back-azimuth. Bin and summing the azimuth is Ok when coverage is similar, but the stations used here have a different number of (high-quality) waveforms. We prefer to keep this format. However, we significantly improved the quality and readability of the figures.

**APPENDIX.**

In order to justify our choice for a compositional parametrization in terms of silica content, we show here the (alpha) quartz abundance related to the composition with different silica content given in table 2 and used in Figure 11, that show a drastic change. We also show similar figures related to the same compositions but with ~1.0 weight percent of water instead than 0.25.

[Figure]

Figure 1: Quartz (alpha) content in vol % as function of pressure and temperature for the seven chemical compositions at variable silica content of table 2 and used in Figure 11. All chemical compositions have 0.25 wt% of water.

[Figure]

*Figure 2: Quartz (alpha) abundance in vol % as function of pressure and temperature for the same chemical compositions as previous figure but with more water content (~1 wt %).*